# Vision-Based Distance Measurement in Advanced Driving Assistance Systems

**Meng Ding [1,2,*], Zhenzhen Zhang [1], Xinyan Jiang [1] and Yunfeng Cao [3]**

[1]  College of Civil Aviation, Nanjing University of Aeronautics and Astronautics, Nanjing 211106, China; zhangzhenzhen@nuaa.edu.cn (Z.Z.); jiangxy@nuaa.edu.cn (X.J.)

[2]  Civil Aviation Key Laboratory of Aircraft Health Monitoring and Intelligent Maintenance, Civil Aviation Administration of China, Nanjing 210016, China

[3]  College of Astronautics, Nanjing University of Aeronautics and Astronautics, Nanjing 210016, China; cyfac@nuaa.edu.cn

*  Correspondence: nuaa_dm@nuaa.edu.cn

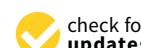

**Featured Application: The outcome of this paper is a deep-learning-based application for the distance measurement between the subject vehicle and the target vehicle or pedestrian, which uses the forward-looking image captured by a vehicle-mounted vision sensor to achieve effective performances of depth map estimation and distance measurement. The technique can be used in advanced driving assistance systems to further enhance driving safety.**

**Abstract:** As the forward-looking depth information plays a considerable role in advanced driving assistance systems, in this paper, we first propose a method of depth map estimation based on semi-supervised learning, which uses the left and right views of binocular vision and sparse depth values as inputs to train a deep learning network with an encoding–decoding structure. Compared with unsupervised networks without sparse depth labels, the proposed semi-supervised network improves the estimation accuracy of depth maps. Secondly, this paper combines the estimated depth map with the results of instance segmentation to measure the distance between the subject vehicle and the target vehicle or pedestrian. Specifically, for measuring the distance between the subject vehicle and a pedestrian, this paper proposes a depth histogram-based method that calculates the average depth values of all pixels whose depth values are in the peak range of the depth histogram of this pedestrian. To measure the distance between the subject vehicle and the target vehicle, this paper proposes a method that first fits a 3-D plane based on the locations of target points in the camera body coordinate using RANSAC (RANdom SAmple Consensus), it then projects all the pixels of the target to this plane, and finally uses the minimum depth value of these projected points to calculate the distance to the target vehicle. The results of the quantitative and qualitative comparisons on the KITTI dataset show that the proposed method can effectively estimate depth maps. The experimental results in real road scenarios and the KITTI dataset confirm the accuracy of the proposed distance measurement methods.

**Keywords:** advanced driving assistance system; semi-supervised network; depth map estimation; distance measurement

---

## 1. Introduction

In order to improve road safety, both the scientific community and manufacturers must pay more attention to the development of automobile safety technology. As one of the key technologies, Advanced Driving Assistance Systems (ADASs) are developing rapidly [1]. Measuring the vehicle–vehicle and vehicle–pedestrian distance is one of the main tasks of

ADASs [2,3]. Generally speaking, existing measurement methods can be placed into two categories [4]: active sensor-based methods and passive vision-based methods. Active sensor-methods, such as LiDAR and ultrasonic sensors, use echo signals to measure the distance to targets. The main advantage of active sensors is that they can be used in different visibility conditions. However, the high cost of the LiDAR system greatly restricts their scope of application [5]. As a result of the lack of shape, texture, and color information, it is difficult to use the data from ultrasonic sensors to collect further useful information from the driving environment. In recent years, because passive vision sensors are relatively cheap and able to capture forward-looking images from the vehicle, containing the rich information required for safe driving, they have become the basic equipment used in existing ADASs [6,7]. The vision-based driving assistance system (V-DAS), a type of ADAS, has been applied in several specific functions of ADAS, such as evasive pedestrian protection, lane-keeping support, and traffic sign warning.

The existing passive vision-based methods used in V-DAS can be further divided into two subcategories [8]: stereo vision-based approaches and monocular vision-based approaches. The former uses multiple view geometry and stereo image pairs to rebuild a 3D space and generate the depth information of the target. However, errors and computational complexities from the calibration and matching of stereo image pairs reduce the measurement accuracy and efficiency to a certain extent in actual road scenarios [9]. Recently, monocular vision-based ADAS has become a research hotspot in the field of intelligent vehicles [10]. Monocular vision methods have certain advantages, such as being cheap, having a simple hardware structure, and a wide field of application. However, because of the lack of a distance scale, traditional monocular vision-based measurement methods cannot complete the absolute distance estimation using only a single image. With the rapid development of deep learning technologies, and using the powerful data mining capabilities of deep learning, it is entirely possible to complete the distance estimation of different targets online based on monocular vision by constructing a deep learning network. Consequently, the main outcome of this paper is to present a system that only uses a single image to complete vehicle–vehicle and pedestrian–vehicle absolute distance measurements online using ADAS. In addition to monocular vision, the proposed system does not need extra sensors as inputs for online applications.

Specifically, in this study, we first constructed and trained a semi-supervised deep learning network which uses the forward-looking visible light image pairs captured by the vehicle-mounted CCD camera and sparse depth labels obtained by LiDAR for offline training. It was then able to estimate the absolute depth value of each pixel of a single image in the camera body coordinate system online. Compared with unsupervised depth estimation methods, the depth estimation results of this semi-supervised network are more accurate. On the basis of the depth map of the input image, focusing on the two main participants in road traffic activities, i.e., pedestrians and vehicles, this paper further proposes a pedestrian–vehicle and a vehicle–vehicle distance measurement method. In summary, the main contributions of this paper are as follows:

(1) This paper combines depth map estimation with Mask-RCNN instance segmentation to propose a vision-based absolute distance estimation method for ADAS. For depth map estimation, we constructed a semi-supervised deep learning network that uses sparse depth labels and left and right views in the offline training process to compute right and left disparity maps, thus improving the performance of offline depth estimation. In the process of online depth map estimation, this deep learning network can convert an RGB (Red, Green, Blue) image into an RGB-D (depth) image, providing depth information corresponding to each pixel of the input image;

(2) To measure the distance between the subject vehicle and the target pedestrian, we propose a depth histogram-based method that calculates the average depth values of all pixels whose depth values are in the peak range of the depth histogram of the pedestrian;

(3) To measure the distance between the subject vehicle and the target vehicle, we propose a method that first fits a 3-D plane based on the locations of target points in the camera body coordinate

using RANSAC, then projects all the pixels of this target to this plane, and finally uses the minimum depth value of these projected points to calculate the distance to the target vehicle.

Although the proposed system needs to use stereo image pairs and sparse depth information to train a semi-supervised network, it is considered to be a monocular vision-based approach because it only needs to input a single image when used online in V-DAS. The experimental results on the public dataset KITTI and in real road scenarios illustrate that the proposed system can use a single vehicle forward-looking image to obtain its corresponding pixel-level depth information and accurately predict the distances to different targets to meet the needs of ADAS.

The remainder of this paper is organized as follows: in Section 2, the related works are briefly discussed. Section 3 presents the details of the proposed depth map estimation method. Section 4 presents the two distance measurement methods based on a forward-looking image and the corresponding depth map. Section 5 gives the flow chart of the proposed distance measurement system, which combines a pretrained semi-supervised network for depth map estimation and the Mask R-CNN network for instance segmentation. Section 6 describes and analyzes the experimental results. The conclusions and future work are presented and discussed in the final section.

## 2. Related Work

Generally speaking, stereo-vision methods are not suitable for distance estimation in ADAS. There are two reasons for this: firstly, these methods are very susceptible to errors in feature extraction and matching. Secondly, they can only achieve relatively sparse and local depth values. Therefore, it is difficult to compute the distances of multiple different targets at the same time through these depth values. Therefore, in this section, we mainly discuss the distance measurement methods based on monocular vision and the progress of the related key technologies.

Currently, monocular-vision methods for distance estimation used in ADAS can be divided into two categories. The first category is based on the geometric relationship and camera imaging model. In these types of methods, several parameters from the camera (e.g., the azimuth and elevation angles of the camera) and the measured object (e.g., width of the target vehicle) need to be provided in advance. Liu et al. used the geometric positional relationship of a vehicle in the camera coordinate system to construct the correspondence between the key points in the world coordinate system and the image coordinate system, and then established a ranging model to estimate the target vehicle distance [11]. Kim et al. used the camera imaging model and the width of the target vehicle to estimate the distance to a moving vehicle that is far ahead [12]. The main disadvantage of such methods is that the accuracy of distance estimation depends heavily on the measurement accuracy of the parameters of the camera or the measured object. The second category involves constructing a regression model using machine learning. Wongsaree et al. trained a regression model using the correspondence between different positions in an image and their corresponding distances to complete distance estimation [13]. Gökçe et al. used the target vehicle information to train a distance regression model for distance estimation [14]. The main disadvantage of these methods is that they have to collect a large number of training data with real distances.

In the proposed method, the first and core task is to complete the depth map estimation. Traditional methods of vision-based depth map estimation are mostly realized using geometric constraints and handcrafted features (e.g., SIFT), such as Structure from Motion (SFM) [15]. The main disadvantage of these traditional methods is that they are very susceptible to errors in feature extraction and matching, and can only achieve relatively sparse and local depth maps. It is difficult to compute the distances of multiple different targets at the same time through these depth maps. In recent years, deep learning-based methods represented by convolutional neural networks (CNN) have been developed in various fields of computer vision [16], and several CNN-based approaches for depth map estimation have been studied [17,18]. Depending on whether they use real depth data as the labels during the training process, these methods can be divided into three categories: supervised, semi-supervised, and unsupervised methods. Moreover, depth values obtained by

vision-based methods can be divided into the absolute depth, which denotes the true depth value of each pixel in the camera coordinate system, and the relative depth, which indicates the relative distance relationship of different pixels in the image.

As one of the representatives of the supervised methods, the Coarse-Fine method [19], proposed by Eigen et al., contains two CNNs of different scales: the coarse-scale CNN, used to estimate the globe depth of the input image; the fine-scale CNN, for optimizing the local details. On the basis of this method, Eigen et al. further proposed a multiscale network architecture [20], which can complete three tasks, including depth estimation, plan normal measurement, and semantic segmentation. Li et al. proposed a combination method in which a CNN is used to regress the depth of superpixels, and a conditional random field is used for post-processing [21]. The supervised methods require the real depth value of each pixel in the input image as the training labels, which are difficult to obtain. Therefore, as a result of the lack of sufficient training samples to train the supervised network, it is difficult for these methods to become popularly adopted in different application fields, such as V-DAS.

Unsupervised methods are generally divided into two subcategories. One is referred to as the self-supervised method, which uses the temporal information from a monocular video as supervision information. Compared with supervised methods, the training samples for a self-supervised network can be easily obtained. However, self-supervised methods also have some shortcomings. Firstly, self-supervised methods have to complete pose estimation using other approaches that increase the complexity of these methods, as a result the depth estimation results are largely dependent on the accuracy of pose estimation. Secondly, because of the lack of scale information, these self-supervised methods can only obtain relative depth results, and cannot obtain the absolute depth values. This relative depth information does not meet the requirements of ADAS. The other subcategory, unsupervised methods, is based on the spatial constraint relationship from the stereo vision acting as the supervision information. This means that stereo vision is used during offline training and monocular vision is used to estimate depth maps online. Generally, since the relative pose of two cameras is known, the estimation results of this subcategory are better than the results of a self-supervised network. Moreover, different from self-supervised networks, because the relative location of two cameras is known, unsupervised methods can obtain the absolute depth value, which is very important for ADAS. Garg et al. first used the spatial constraint of two views to propose a depth estimation method for unsupervised monocular vision based on convolutional neural networks [22]. Garg's method utilizes a network structure similar to a full convolutional neural network (FCN), including encoding and decoding. In the unsupervised network, the depth map is first obtained by inputting the left view into the CNN, and then the corresponding disparity map is calculated according to the relationship between the disparity and the depth in the stereo vision. Furthermore, both this disparity map and the right view are used to reconstruct the left view; the error between the original left view and reconstructed left view is used as the loss function of the encoding–decoding network. On the basis of this network structure, Godard et al. proposed a loss function that contains appearance matching loss, disparity smoothness loss, and left-right disparity consistency loss [23]. However, the estimation accuracy of unsupervised methods would be further improved if new information, including real depths, was added to the loss functions during training.

Generally speaking, compared to dense depth information corresponding to each pixel of a forward-looking image, sparse depth information corresponding to parts of pixels is easier to obtain. Therefore, semi-supervised methods using sparse and local depth information have been recently studied. Kuzniestsov et al. combined a sparse ground truth depth map with a calibrated stereo image pair to train a semi-supervised network [24], which demonstrated state-of-the-art performance using the KITTI dataset. Moreover, Ji et al. proposed a novel semi-supervised adversarial learning framework that only utilizes a small number of image-depth pairs in conjunction with a large number of easily available monocular images to achieve depth estimation [25]. In summary, compared with unsupervised methods, semi-supervised methods can achieve better estimation results due to the introduction of local and sparse depth labels.

In this study, we used instance segmentation to build a bridge between the depth map estimation and the distance measurement of a specific object. Instance segmentation is based on object detection and semantic segmentation, providing different labels for separate instances of objects belonging to the same class. As a relatively flexible model for instance segmentation, Mask R-CNN inherits the basic framework of Faster R-CNN and adds an object mask prediction branch [26]. As Mask R-CNN is easy to transfer to other tasks, is superior to most methods of instance segmentation, and only increases the computational load slightly as compared to Faster R-CNN, this paper employs Mask R-CNN to segment the target from the background for target distance estimation. One of the earliest applications of instance segmentation for distance estimation was performed by Huang et al. They proposed a method that combines instance segmentation and a projection geometry model for distance estimation [27]. In the latest work of Huang et al., they obtained the vehicle attitude angle using an angle regression model and a segmentation algorithm, and then estimated the distance to the vehicle ahead by constructing an "area-distance" geometric model [4].

In this paper, we combine the results of depth map estimation, which provide the depth information of each pixel, with the results of instance segmentation, which provide the classification information of each pixel, to estimate the absolute distances to different participants on the road, e.g., cars, vans, trucks, and pedestrians.

## 3. Depth Map Estimation

### 3.1. Relationship between Disparity and Depth

The depth estimation principle based on the left and right views is shown in Figure 1, where one image pair contains two images $\mathbf{I}^l$ and $\mathbf{I}^r$ captured simultaneously by the left and right cameras; $f$ and $b$ are the focal length and baseline distance, respectively, $P_l$ and $P_r$ are the projection points of the object point $P$ on the imaging planes of the left and right cameras, respectively, and $(x_l, y_l, f)$ and $(x_r, y_r, f)$ are the locations of $P_l$ and $P_r$ in the coordinate systems of the left and right camera body, respectively. To simplify the following description, we set $y_l = 0$ and $y_r = 0$. As shown in Figure 1, according to the property of similar triangles,

$$\frac{x_p^l}{x_l} = \frac{z_p^l}{f}, \frac{x_p^r}{x_r} = \frac{z_p^r}{f} \tag{1}$$

where $(x_p^l, z_p^l)$ and $(x_p^r, z_p^r)$ are the locations of $P$ in the coordinate systems of the left and right camera bodies, respectively. As $x_p^l - x_p^r = b$ and $z_p^r = z_p^l$, as shown in Figure 1, Equation (1) can be rewritten as follows:

$$\frac{x_p^l}{x_l} = \frac{z_p^l}{f}, \frac{x_p^l - b}{x_r} = \frac{z_p^l}{f}, \tag{2}$$

further,

$$z_p^l = \frac{fb}{d} \tag{3}$$

where $d = x_l - x_r$ is the left–right disparity and represents the difference in the location of point $P$ in the left and right images. From Equation (3), we find that $z_p^l$, denoting the depth of point $P$ in the coordinate system of the left camera body, can be obtained if the baseline distance $b$, the camera focal length $f$, and the disparity $d$ are all known. Therefore, depth estimation can be transformed into a problem of solving disparity map computation. Consequently, the main task of a deep learning network is to compute the disparity map from the input image pair.

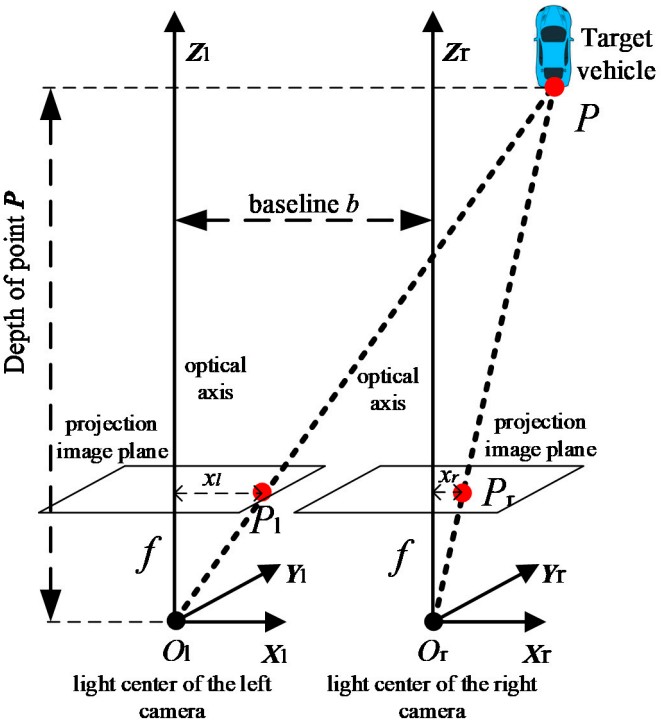

**Figure 1.** The depth estimation principle based on the left and right views.

### 3.2. Semi-Supervised Learning Network for Depth Map Estimation

Figure 2 shows the training processing of a semi-supervised learning network for depth map estimation. During the training process, the inputs are the left and right views, and the corresponding sparse depth labels that have been matched with the left and right views, respectively. The outputs of the deep network are two disparity maps corresponding to the left and right views, respectively. The loss functions used for training this network contain appearance matching loss, disparity smoothness loss, left–right disparity consistency loss, and supervised loss.

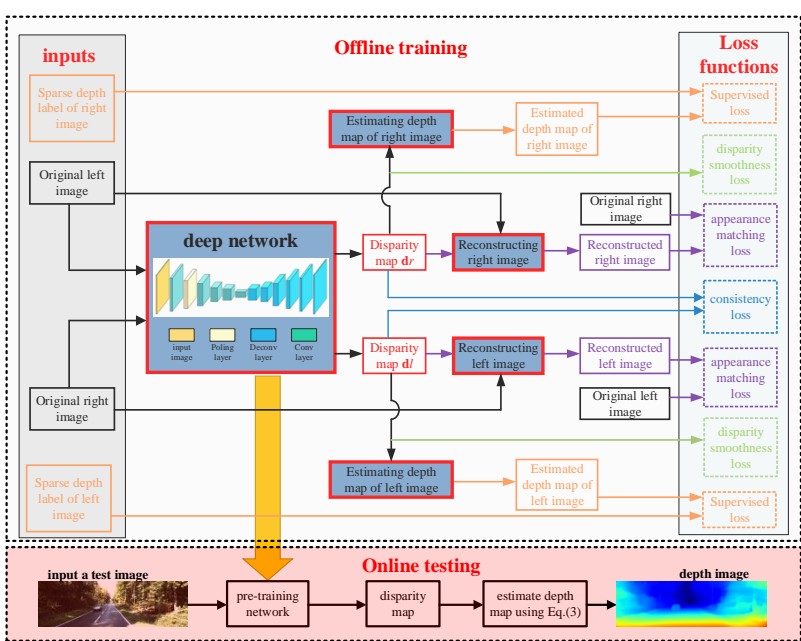

**Figure 2.** Depth estimation network and its loss functions.

As shown in Figure 2, the semi-supervised depth estimation network in this paper is based on the encoding–decoding network structure. It uses ResNet-50 as the feature extraction model in the encoding stage [28], and the ResNet-50 network in the opposite direction in the decoding stage.

### 3.2.1. Loss Functions

Appearance Matching Loss

In the training process, the left view $\mathbf{I}^l$ and right view $\mathbf{I}^r$ captured by the left and right cameras, respectively, are simultaneously input into the network. When the left view $\mathbf{I}^l$ is input into this depth estimation network, the disparity map $\mathbf{d}^r$ that corresponds to the conversion from the left view to the right view can be predicted pixel by pixel. Similarly, the disparity map $\mathbf{d}^l$ is obtained when the input image is the right view $\mathbf{I}^r$. Further, we can reconstruct the right view $\tilde{\mathbf{I}}^r$ based on the original left view $\mathbf{I}^l$ and the right disparity map $\mathbf{d}^r$, and the left view $\tilde{\mathbf{I}}^l$ based on the right view $\mathbf{I}^r$ and the left disparity map $\mathbf{d}^l$. Finally, the reconstructed left view $\tilde{\mathbf{I}}^l$ and right view $\tilde{\mathbf{I}}^r$ are respectively combined with the original left view $\mathbf{I}^l$ and right view $\mathbf{I}^r$ to form a loss function, known as the appearance matching loss for network training. This appearance matching loss is expressed as follows [29]:

$$
\begin{aligned}
C_{ap}^l &= \frac{\alpha}{mn}\sum_{i=1}^{m}\sum_{j=1}^{n}\frac{1-X_{\text{SSIM}}(\mathbf{I}^l(i,j),\tilde{\mathbf{I}}^l(i,j))}{2} + (1-\alpha)\|\mathbf{I}^l(i,j)-\tilde{\mathbf{I}}^l(i,j)\|_1 \\
C_{ap}^r &= \frac{\alpha}{mn}\sum_{i=1}^{m}\sum_{j=1}^{n}\frac{1-X_{\text{SSIM}}(\mathbf{I}^r(i,j),\tilde{\mathbf{I}}^r(i,j))}{2} + (1-\alpha)\|\mathbf{I}^r(i,j)-\tilde{\mathbf{I}}^r(i,j)\|_1
\end{aligned}
\tag{4}
$$

where $\mathbf{I}^l, \mathbf{I}^r, \tilde{\mathbf{I}}^l$ and $\tilde{\mathbf{I}}^r \in \mathbb{R}^{m\times n}$, $\alpha = 0.85$, $X_{\text{SSIM}}(\mathbf{x},\mathbf{y})$ can be calculated using Ref. [30].

Disparity Smoothness Loss

Disparity smoothness loss consists of two parts: (1) the gradient values of the disparity map in two directions $\left|\mathbf{d}^l(i+1,j)-\mathbf{d}^l(i-1,j)\right|$ and $\left|\mathbf{d}^l(i,j+1)-\mathbf{d}^l(i,j-1)\right|$ are used to create local smoothness; (2) considering that depth discontinuities often occur at image edges, we weight the gradient values of the disparity map with an edge-aware term using the image gradients in two directions.

$$
\begin{aligned}
C_{ds}^l &= \frac{1}{mn}\sum_{i=1}^{m}\sum_{j=1}^{n}\left|\mathbf{d}^l(i+1,j)-\mathbf{d}^l(i-1,j)\right|\exp\!\left(-\left|\mathbf{I}^l(i+1,j)-\mathbf{I}^l(i-1,j)\right|\right)+ \\
&\quad \left|\mathbf{d}^l(i,j+1)-\mathbf{d}^l(i,j-1)\right|\exp\!\left(-\left|\mathbf{I}^l(i,j+1)-\mathbf{I}^l(i,j-1)\right|\right) \\
C_{ds}^r &= \frac{1}{mn}\sum_{i=1}^{m}\sum_{j=1}^{n}\left|\mathbf{d}^r(i+1,j)-\mathbf{d}^r(i-1,j)\right|\exp\!\left(-\left|\mathbf{I}^r(i+1,j)-\mathbf{I}^r(i-1,j)\right|\right)+ \\
&\quad \left|\mathbf{d}^r(i,j+1)-\mathbf{d}^r(i,j-1)\right|\exp\!\left(-\left|\mathbf{I}^r(i,j+1)-\mathbf{I}^r(i,j-1)\right|\right)
\end{aligned}
\tag{5}
$$

Left–Right Disparity Consistency Loss

In order to achieve the consistency of left and right disparity maps, the left–right disparity consistency loss is introduced to make the left-view disparity map equal to the projected right-view disparity map. This loss is shown as follows:

$$
\begin{aligned}
C_{sv}^l &= \frac{1}{N}\sum_{(i,j)\in\Omega}\|\tilde{\mathbf{Z}}^l(i,j)-\mathbf{Z}^l(i,j)\|_\delta \\
C_{sv}^r &= \frac{1}{N}\sum_{(i,j)\in\Omega}\|\tilde{\mathbf{Z}}^r(i,j)-\mathbf{Z}^r(i,j)\|_\delta
\end{aligned}
\tag{6}
$$

Supervised Loss

The main difference between unsupervised and semi-supervised depth estimation is that the semi-supervised method adds a supervised loss to the above three losses. The prerequisite for using supervised loss is to know the true depth values and the predicted depth values. In the training process, the true depth values corresponding to parts of pixels are first obtained and matched. The predicted depth values of the pixels with true depth values can be converted from the predicted disparity map using Equation (3). The supervised loss can be defined as the deviation of the predicted depth values $\tilde{\mathbf{Z}}$ from the available ground truth $\mathbf{Z}$, and expressed as follows:

$$
\begin{aligned}
C_{sv}^l &= \frac{1}{N} \sum_{(i,j) \in \Omega} \|\tilde{\mathbf{Z}}^l(i,j) - \mathbf{Z}^l(i,j)\|_\delta \\
C_{sv}^r &= \frac{1}{N} \sum_{(i,j) \in \Omega} \|\tilde{\mathbf{Z}}^r(i,j) - \mathbf{Z}^r(i,j)\|_\delta
\end{aligned}
\tag{7}
$$

where $\Omega$ is the set of all pixels with true depth values, and $N$ is the number of the pixels with true depth values. $\|\bullet\|_\delta$ is the berHu norm and defined as follows:

$$
\|d\|_\delta = \begin{cases} |d|, d \le \delta \\ \frac{d^2 + \delta^2}{2\delta}, d > \delta \end{cases}
\tag{8}
$$

and

$$
\delta = 0.2 \max_{(i,j) \in \Omega_Z} \left( \left| \tilde{\mathbf{Z}}(i,j) - \mathbf{Z}(i,j) \right| \right)
\tag{9}
$$

Loss Function for Depth Estimation

The total loss function for depth estimation consists of four parts: appearance matching loss, disparity smoothness loss, left–right disparity consistency loss, and supervised loss. As a combination of four loss functions, the expression is as follows:

$$
C_z = \left( C_{ap}^l + C_{ap}^r \right) + \left( C_{ds}^l + C_{ds}^r \right) + \left( C_{lr}^l + C_{lr}^r \right) + \left( C_{sv}^l + C_{sv}^r \right)
\tag{10}
$$

This paper presents a semi-supervised method that adds a supervised loss item to the unsupervised method to complete the depth estimation. Compared with unsupervised methods that only use the left and right views, this semi-supervised method can improve the estimation accuracy by introducing sparse and local depths corresponding to parts of pixels. Since the resolution and scan range are limited, 3-D LiDAR can only scan some points corresponding to parts of the image pixels and obtain sparse depth information of the front view scene of the vehicle.

During the training process, in order to make full use of the sparse and local depth information, we increase the use of the supervised loss function in the unsupervised framework. Specifically, for pixels with true depth values, we employ this depth information as the ground truth and add a supervised loss function. Additionally, for pixels without depth labels, the unsupervised method based on the principle of binocular reconstruction is used. This combination of unsupervised and supervised methods is referred to as semi-supervised depth estimation.

### 3.2.2. Depth Map Estimation

When the offline training is completed, we can obtain a pretrained depth estimation network. In the online test process, only one single test image is inputted into this pretrained depth estimation network, and the disparity map corresponding to this input image is calculated. Finally, according to Equation (3), the depth value of each pixel of the input image in the coordinate system of the left camera body can be estimated by combining the focal length and baseline distance of the binocular camera used for training. The depth values of all pixels form a depth map corresponding to the input image.

## 4. Distance Measurement between the Target and the Subject Vehicle

### 4.1. Pixel-Level Depth Map of the Target

Using the trained semi-supervised depth estimation network, we can obtain the pixel-level depth map. In order to measure the distance between the target and the subject vehicle, it is necessary to detect the pixels belonging to the target from the input forward-looking image. It is well known that instance segmentation can achieve pixel-level target classification. As a general instance segmentation architecture, Mask R-CNN is based on the Faster R-CNN detector and identifies the pixel-level regions of the target by adding a branch for the segmentation task. According to the results of instance segmentation, we can obtain the depth value of each pixel of the target. Figure 3 shows two forward-looking images captured by ADAS. The left image contains a car, the pixels of this car from instance segmentation, and the corresponding depth maps in 2D and 3D spaces, respectively. The right image in Figure 3 contains a pedestrian, the pixels of this pedestrian from instance segmentation, and the corresponding depth maps in 2D and 3D spaces, respectively.

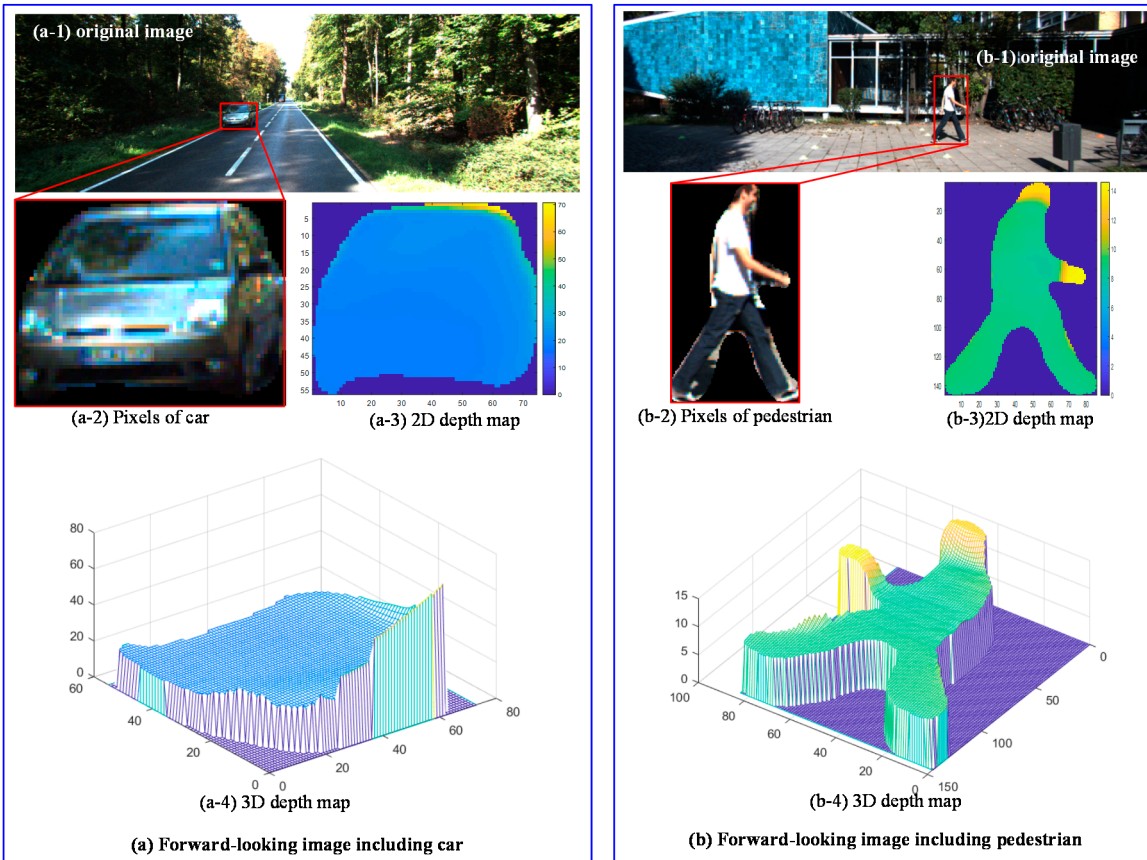

**Figure 3.** Pixel-level depth maps of the target. (**a**) Forward-looking image including a car and its depth map. (**b**) Forward-looking image including a pedestrian and its depth map.

### 4.2. Target Distance Measurement

Figure 4 shows the pixel depth values of the car and pedestrians in Figure 3 in the camera body coordinate system. Generally speaking, to ensure a certain safety margin of ADSA, the minimum depth value in all pixels of this target can be regarded as the distance between the target and subject vehicle. However, in order to reduce the influence of the noise and error of depth map estimation, we present different methods to measure this distance according to different objects. As we all know, the three-dimensional structure of a vehicle can simply be considered as composed of multiple planes. On the contrary, the shape of the human body is a curved surface. The above spatial structures and

shapes of the vehicle and the pedestrian can be observed from Figure 4a,b. Therefore, in the proposed methods, we divide the road targets into two types: vehicle (e.g., car, van, truck) and pedestrian, based on their 3D shapes, and we use two approaches to measure the distance, respectively.

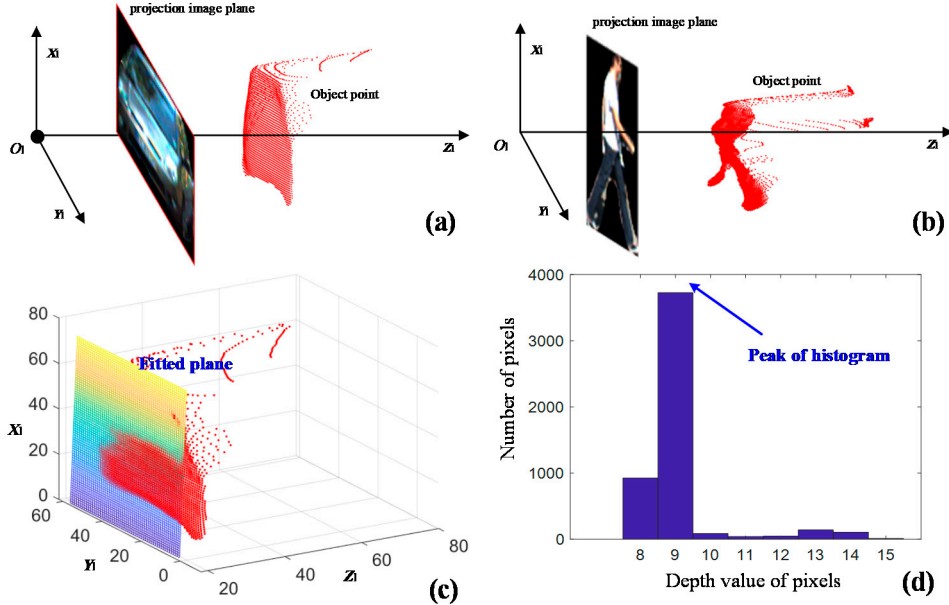

**Figure 4.** Locations of the object points in the camera body coordinate system. (**a**) The object point locations of the car. (**b**) The object point locations of the pedestrian. (**c**) The fitted plane using the object points of the car. (**d**) The depth histogram of the object points of the pedestrian.

### 4.2.1. Distance Measurement of the Target Vehicle

If the target is a vehicle, we first fit a plane in the camera body coordinate system using object points corresponding to the pixels of the vehicle and the RANSAC algorithm [31]. Suppose that $\left(x_I^i, y_I^i, z_I^i\right)$ is the coordinate of the point corresponding to each pixel of the target vehicle in the camera body coordinate system, $S$ is the number of these object points, and $i = 1, \ldots, S$. By using RANSAC, a fitted plane in the camera body coordinate system can be determined and expressed as $z = ax + by + c$. Secondly, by projecting the image points $\left(x_I^i, y_I^i, z_I^i\right)$ onto this plane, specifically, $\hat{z}_I^i = ax_I^i + by_I^i + c$, we can obtain a set $\hat{Z} = \left\{\hat{z}_I^i \middle| i = 1, 2, \ldots, S\right\}$. Finally, the distance between the target vehicle and the subject vehicle is equal to the minimum value of this set $\hat{Z}$.

### 4.2.2. Distance Measurement of the Target Pedestrian

Different from a target vehicle, the object points of the pedestrian in the camera body coordinate system cannot be fitted as a plane. Consequently, this method first uses the histogram to count the number of object points with different depth values. Specifically, suppose that $\left(x_I^i, y_I^i, z_I^i\right)$ is the coordinate of the object points of the target pedestrian, $S$ is the number of object points, *min_z* and *max_z* are the minimum and maximum depth values of the object points, respectively, the statistical range of depth values in the histogram is from $\lfloor min\_z \rfloor$ to $\lceil max\_z \rceil$, and the interval is 1, where $\lfloor \bullet \rfloor$ and $\lceil \bullet \rceil$ indicate rounding down and rounding up to an integer, respectively. Secondly, the peak of the histogram and the corresponding depth range are obtained. Finally, the distance between the target pedestrian and the subject vehicle is the average of the depth values in this range.

### 5. Proposed Method Implementation

On the basis of the pretrained semi-supervised network for depth map estimation and the Mask R-CNN network for instance segmentation, the flow chart of the proposed distance measurement method is as follow (shown in Figure 5).

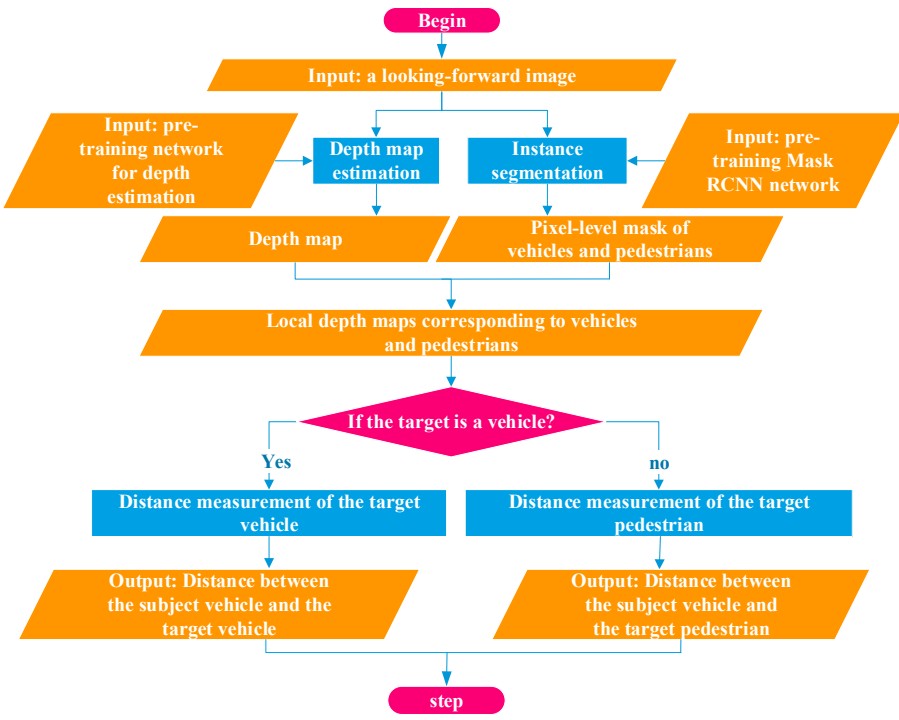

**Figure 5.** Flow chart of the proposed distance measurement method.

It is important to note that when the binocular vision equipment used in offline training is different from the image sensor used in the online test, in order to obtain the absolute depth value, it is necessary to adjust and calibrate the focal length of the online sensor based on the image resolution and focal length of the training equipment. The relationship between two focal lengths is as follows:

$$f_{test} = \frac{f_{train} w_{train}}{w_{test}} \tag{11}$$

where $f_{train}$ and $f_{test}$ are the focal lengths of sensors for training and testing, respectively, and $w_{train}$ and $w_{test}$ are the widths of the training and test images, respectively. As mentioned above, the output of the pretrained network for depth estimation is the disparity map **d**. In order to obtain the depth value of each pixel of the input image, we must use the following equation:

$$\mathbf{D}(i,j) = \frac{f_{test} b}{\mathbf{d}(i,j)} \tag{12}$$

where $\mathbf{D}(i,j)$ is the depth value of each pixel, $\mathbf{d}(i,j)$ is the disparity value of each pixel, and $b$ is the baseline length of the binocular vision equipment used in offline training.

## 6. Experiments and Results

### 6.1. Implementation Details

Firstly, we trained a semi-supervised network for depth estimation on a computation hardware platform with NVIDIA GeForce GTX 1080Ti (NVIDIA, Santa Clara, CA, USA), with the Ubuntu 14.04 (Canonical, London, UK) operating system and the TensorFlow1.4.0 (Google, Mountain View, CA, USA) development tool. In the KITTI dataset, we selected 7322 groups as the training set, in which each group contained right and left views and two corresponding sparse depth maps [32]. KITTI is a popular dataset which can be used for vision algorithm testing of ADAS; it contains a large number of stereo image pairs captured from a car driving in an urban scenario and also provides sparse depth data matched with the stereo vision. These depth data were not only the sparse depth labels in the

training process, but also the ground truth for algorithm evaluation. During depth estimation network training, we used stochastic gradient descent with an initial learning rate of 0.0001 and 50 epochs. From the 30th to 40th epoch, the learning rate was reduced to 1/2 of the initial value, and the learning rate of the last 10 epochs was reduced to 1/4 of the initial value. The batch size was equal to 8. We used the Adam optimizer to optimize the model, and set $\beta_1 = 0.9$ and $\beta_2 = 0.999$. The Mask R-CNN model used in this paper was downloaded from https://github.com/matterport/Mask_RCNN.

In order to assess the performance of depth map estimation, which is the key of distance measurements, we used the following depth evaluation metrics [33]:

(1) Absolute relative error (*AbsRel*)

$$AbsRel = \frac{1}{N} \sum_{i=1}^{N} \frac{|\widetilde{z}_i - z_i|}{z_i} \tag{13}$$

(2) Root-mean-square error (*RMSE*)

$$RMSE = \sqrt{\frac{1}{N} \sum_{i=1}^{N} \|\widetilde{z}_i - z_i\|^2} \tag{14}$$

(3) Threshold accuracy

$$\max\left(\frac{\widetilde{z}_i}{z_i}, \frac{z_i}{\widetilde{z}_i}\right) = \delta \tag{15}$$

where the threshold usually takes three values: $1.25$, $1.25^2$, and $1.25^3$; for different thresholds, there are different threshold accuracies: $\delta < 1.25$, $\delta < 1.25^2$, and $\delta < 1.25^3$; $N$ is the number of pixels with ground truth in the test set; $\widetilde{z}_i$ and $z_i$ are the predicted depth value and true depth value, respectively. Regarding the above evaluation indexes, the smaller the first two parameters (*AbsRel* and *RMSE*), the higher the accuracy of the depth estimation result. Conversely, the larger the threshold accuracy, the better the depth estimation result.

## 6.2. Performance Comparison of Depth Estimation

### 6.2.1. Quantitative Comparison with the Other Four Methods

In this subsection, we provide a comprehensive comparison of the proposed depth map estimation method with four other methods: Eigen's method [20], which is a supervised depth estimation method; Zhou's [34] and Godard's [23] methods, which are unsupervised estimation methods; Kuznietsov's [24] method, which is a semi-supervised method. We conducted the quantitative comparison experiments on the KITTI dataset. We used 200 images from the KITTI 2015 dataset as the test samples. In Table 1, we illustrate the comparative results of the proposed method with the other four methods on the KITTI dataset. We found that in terms of absolute relative error, the semi-supervised method presented in this paper is a huge improvement as compared with the supervised method of Eigen et al. and the unsupervised method of Zhou et al. The results obtained by our method are also significantly better than the results of Godard's method which uses left–right consistency. In addition, our method outperforms the Kuznietsov's method which is also a semi-supervised method. From the perspective of the root-mean-square error, the method in this paper is significantly improved compared to the other four methods. In terms of threshold accuracy, accuracies of more than 90% were achieved for the three thresholds of our method, all being higher than the other methods.

**Table 1.** Quantitative comparison on the KITTI dataset.

| Method | Category | Abs_Rel | RMSE | $\delta < 1.25$ | $\delta < 1.25^2$ | $\delta < 1.25^3$ |
|---|---|---|---|---|---|---|
| Eigen et al. | Supervised | 0.203 | 6.307 | 0.702 | 0.890 | 0.958 |
| Zhou et al. | Unsupervised | 0.183 | 6.709 | 0.734 | 0.902 | 0.959 |
| Godard et al. | Unsupervised | 0.128 | 5.547 | 0.815 | 0.922 | 0.968 |
| Kuznietsov et al. | Semi-supervised | 0.076 | 3.842 | 0.903 | 0.948 | 0.975 |
| Ours | Semi-supervised | **0.071** | **3.740** | **0.934** | **0.979** | **0.992** |

### 6.2.2. Ablation Study of Depth Map Estimation

Compared with unsupervised networks, semi-supervised networks can improve the performance of depth map estimation by introducing the sparse and local depth labels. In this subsection, we present our experiment to demonstrate the function of these sparse labels. Figure 6 shows the comparative results of the unsupervised and semi-supervised methods. The first row contains three forward-looking images captured by on-board vision system and containing vehicles and pedestrians. Three maps of the 3-D point cloud are the sparse and local depth labels corresponding to the three images in the first row. From the depth labels in the second row, we can observe that the ground truth of the depth value is mainly concentrated in the middle area of each image, and there is no ground truth in the upper and lower edges. The third row contains the depth map estimation results obtained by the unsupervised method, which only uses left–right consistency. The last row contains the depth maps estimated by the proposed semi-supervised method. By comparing the areas in white rectangles in the third and fourth rows, and considering the areas in red rectangles in the first row, we can see that the results of the semi-supervised method can estimate depth values more accurately. For example, the depth prediction of strip objects such as pedestrians and traffic signs is more accurate.

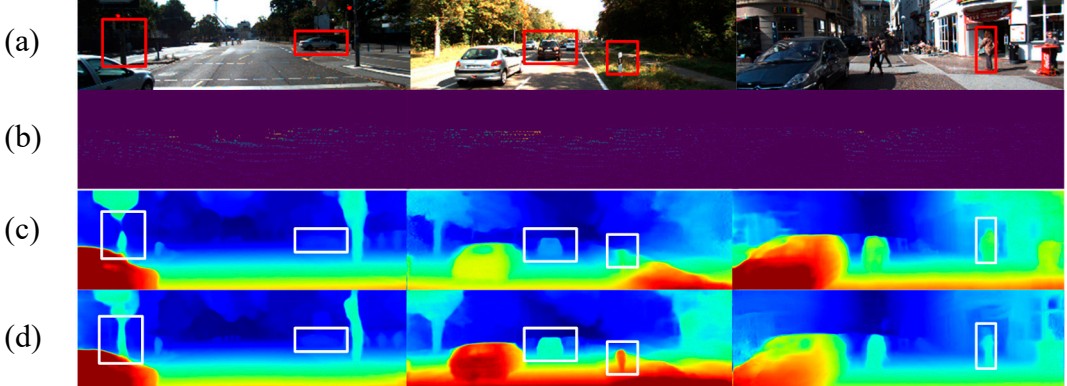

**Figure 6.** Qualitative comparison of unsupervised and semi-supervised depth map estimation. (**a**) The first row shows the forward-looking images used as the inputs of the depth map estimation network. (**b**) The second row shows the sparse and local depth values as the training labels. (**c**) The third row shows the results of the unsupervised method. (**d**) The fourth row shows the results of our method.

### 6.2.3. Depth Map Estimation in Real Road Scenarios

In order to further verify the generalization ability of the proposed method, we directly used the depth estimation model trained on the KITTI dataset to do tests in real road scenarios. The results are shown in Figure 7 and clearly show that even if there are no road driving scene images with similar perspectives added to the training sample set, due to its good generalization ability, the proposed depth estimation network can still effectively recover depth information from the test scene and meet the requirements of distance estimation in ADAS.

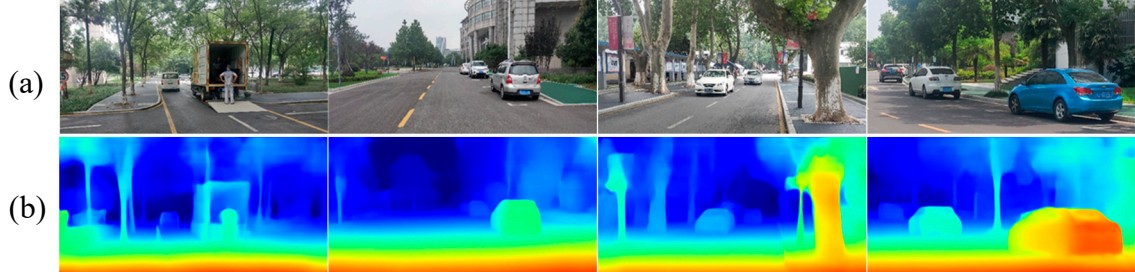

**Figure 7.** Results of depth map estimation in real road scenarios. (**a**) The first row shows forward-looking images captured from the real road scenarios. (**b**) The second row shows the depth maps of the images in the first row using the proposed method of depth map estimation.

## 6.3. Distance Measurement of the Target

### 6.3.1. Distance Measurement of the Pedestrian

In this subsection, we evaluate our distance measurement method for pedestrians. Our method computes the depth average of all pixels whose depth values are in the peak range of the depth histogram. To verify the effectiveness of our measurement method, we compared it with the method that calculates the average depth of all pixels belonging to the pedestrian region. In the experiment, under the premise of fixing the camera position, we arranged the human to stand at the positions $L$ and $2L$, away from the camera position, and conducted quantitative comparative experiments under four different $L$ values, as shown in Figure 8. Figure 8a is a schematic diagram of our experimental device, Figure 8b,c shows four original images corresponding to different $L$ values (2.82, 3.93, 5.87, and 7.86 m) and their depth maps using the proposed depth map estimation network.

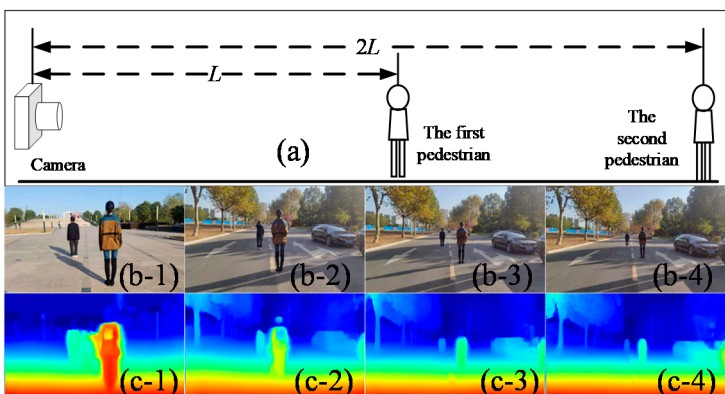

**Figure 8.** Distance measurement of the pedestrian. (**a**) Schematic diagram of the experimental device. (**b**) Four original images corresponding to different $L$ values: $L$ = 2.82 m in (**b-1**); $L$ = 3.93 m in (**b-2**); $L$ = 5.87 m in (**b-3**); $L$ = 7.86 m in (**b-4**). (**c**) Depth maps corresponding to four input images of (**b**) using the proposed depth map estimation network.

The distance measurement results of the pedestrian are given in Table 2. According to these results, we came to the following two conclusions: (1) the pedestrian distance measurement error of our method is significantly smaller than the error obtained using the average depth value; (2) the measurement results of our method effectively reflect the relative distance between two pedestrians, i.e., the distance between the second pedestrian and the camera is twice the distance between the first pedestrian and the camera.

**Table 2.** Results of pedestrian distance measurement.

| L (Ground Truth) | Average Depth Value | | Our Method | |
|---|---|---|---|---|
| | The First Pedestrian | The Second Pedestrian | The First Pedestrian | The Second Pedestrian |
| 2.82 | 2.96 | 5.82 | 2.67 | 5.56 |
| 3.93 | 3.76 | 7.48 | 3.97 | 7.92 |
| 5.87 | 5.44 | 12.07 | 5.82 | 11.34 |
| 7.86 | 7.51 | 15.72 | 8.13 | 16.02 |
| Average error | 0.298 | 0.267 | **0.158** | **0.167** |
| | 0.283 | | **0.162** | |

### 6.3.2. Distance Measurement of the Vehicle

In the vehicle distance measurement experiment, we selected a car and an SUV (as shown in Figure 9), which are common in road scenarios, as the experimental objects, and used a laser rangefinder to measure the minimum horizontal distance to the target, starting from 2.5 m, taking a picture every 2.5 m, and using a camera with the focal length of 4.58 mm to 12.5 m. To test the performance of our vehicle distance measurement method, we compared it with the aforementioned average depth method and the method used in the pedestrian distance measurement. The comparative results are shown in Table 3. From these, we came to the following conclusions: (1) compared with the car distance measurement results, the measurement accuracy for the SUV was better; this is because the rear of an SUV is similar to a plane and so the plane fitting error is smaller; (2) no matter what method was used, as the real distance between the subject camera and target vehicle increased, the measurement accuracy decreased significantly. We believe there are two reasons for this. First, when the distance is greater, the mask of the target becomes smaller and thus the mask error becomes larger, i.e., the pixels that do not belong to the target area are extracted for the distance measurement. Secondly, according to the principle of binocular vision, since the baseline length is fixed, the longer the distance, the lower the measurement accuracy; (3) our vehicle distance measurement method demonstrated a better performance than the other two methods. Specifically, for the SUV distance measurement, the results of our method were better than the other methods for all five distances, and for the car distance measurement, our method was the best for three of the five distances.

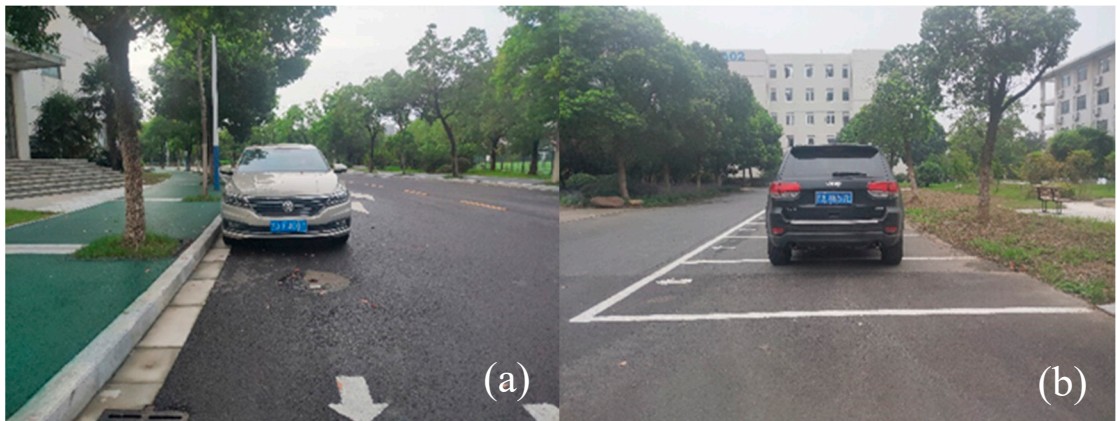

**Figure 9.** Test vehicles containing (**a**) a car and (**b**) an SUV.

**Table 3.** Results of the vehicle distance measurement. (The best result in each row and each type of vehicle is shown in bold and italics).

| Distance (Ground Truth) | Average Depth Value | | Method of Depth Average | | Ours | |
|---|---|---|---|---|---|---|
| | **Car** | **SUV** | **Car** | **SUV** | **Car** | **SUV** |
| 2.5 m | 4.7 | 2.9 | 5.4 | 2.7 | *2.9* | *2.4* |
| 5 m | 7.6 | 6.8 | 5.8 | 5.3 | *5.0* | *5.0* |
| 7.5 m | 11.1 | 8.2 | **7.6** | 7.1 | 7.3 | *7.4* |
| 10 m | **10.6** | 8.8 | 8.5 | 7.7 | 9.2 | *9.3* |
| 12.5 m | 14.3 | *9.9* | 10.4 | 9.5 | **10.8** | *9.9* |

### 6.3.3. Distance Measurement on the KITTI Dataset

In this section, we provide several distance measurement results using the proposed method, which combines semi-supervised depth estimation, Mask RCNN, and pedestrian and vehicle distance measurement capabilities. Figure 10 shows several forward-looking images from the KITTI dataset, the red rectangles represent the targets, i.e., pedestrians (P), trucks (T), vans (V), and cars (C). The true distances of these targets and the corresponding estimated results are shown in Figure 11. In Figure 11, there are 28 targets, containing 3 pedestrians, 1 truck, 3 vans, and 21 cars. The average distance error rate of the 28 targets was 5.56%, in which the average distance error rate of the pedestrians was 4.02%, and the vehicles' average error rate was 5.74%. Compared with the measurement results using images captured by our camera, the accuracy of the distance measurements using the KITTI dataset was obviously higher. This is mainly because our model for depth map estimation used KITTI images as training samples. It is worth noting that the average processing time using the proposed distance measurement method for each image was 1.824 s, of which the online running time for the depth map estimation was 0.085 s, the time required for MASK-RCNN as 1.682 s, and computational time of distance measurement was 0.057 s. Therefore, in order to improve the real-time performance of the proposed method, it is necessary to improve the computational efficiency of the instance segmentation algorithm.

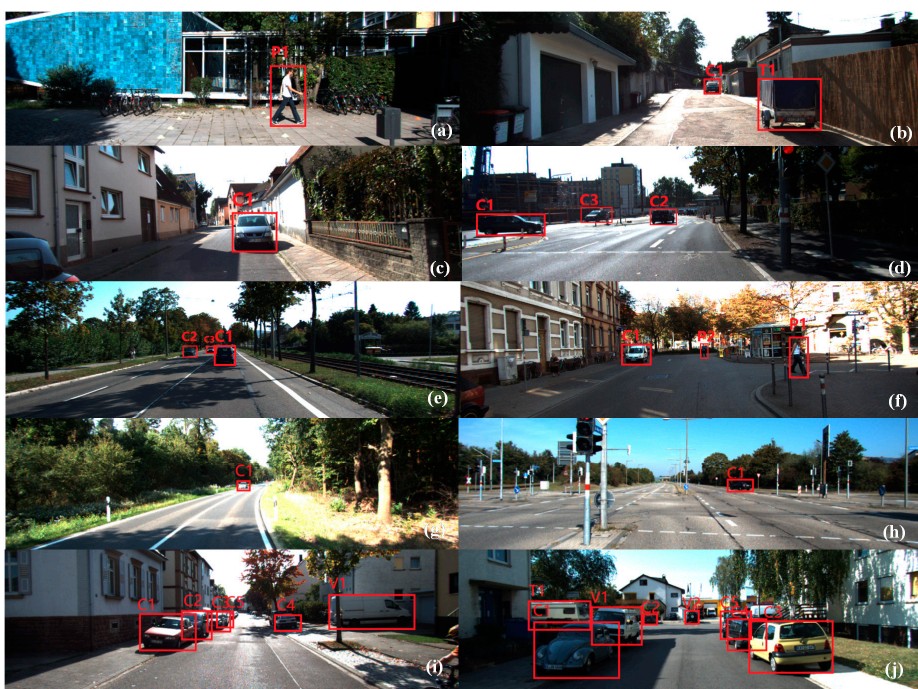

**Figure 10.** Several examples from the KITTI dataset for distance measurement. Images (**a**)–(**j**) are the forward-looking images from the KITTI dataset which show the bounding boxes and categories.

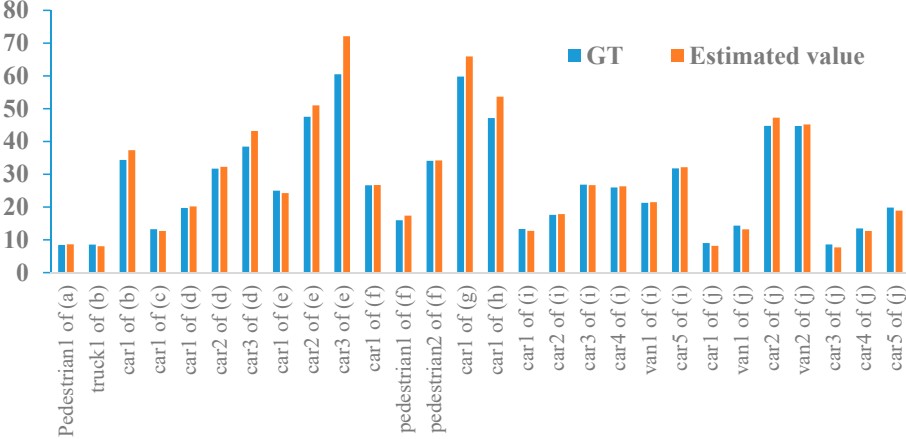

**Figure 11.** Ground truths (GT) of the distance and corresponding estimated values of all targets in Figure 10 using the proposed architecture (unit: m).

We also tested the performance of depth map estimation when the target brightness changed due to the angle of light irradiation. In Figure 12a, as a result of the reflection of light, the brightness values of some areas on the rear windshield of the vehicle in the red rectangle are too large. Conversely, the rear of the vehicle in the red rectangle in Figure 12b is in shadow. Figure 12c,d shows the corresponding depth maps of Figure 12a,b, respectively. From these two depth maps, we can observe that the depth values of pixels in the overexposed areas and shadowy areas have not changed significantly. Therefore, in a daytime road environment, the direction and intensity of light had little effect on the results of depth map estimation and distance measurement using the proposed method.

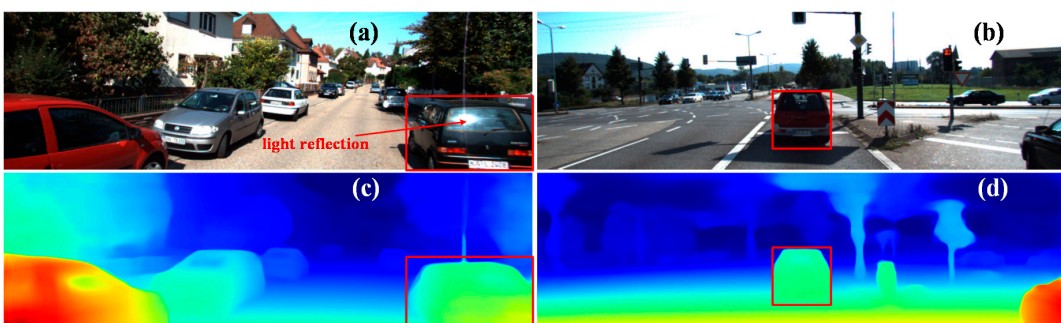

**Figure 12.** The results of depth map estimation in the overexposed area and shadowy area. (**a**) Overexposed area. (**b**) Shadowy area. (**c**) Depth map of (**a**). (**d**) Depth map of (**b**).

## 7. Conclusions

The distance information between the target vehicle or pedestrian and the subject vehicle plays a very important role in ADAS. Therefore, this paper firstly proposed a semi-supervised depth map estimation algorithm, and then combined it with the Mask-RCNN instance segmentation algorithm to propose different distance measurement methods for target pedestrians and target vehicles. The depth map estimation algorithm in this paper used the left and right views of binocular vision and sparse depth ground truth to pretrain an encoding–decoding network. In the process of depth estimation, we used the known camera focal length, baseline length of training samples, and the pretrained deep model to compute the absolute depth map of a single input image. On the basis of the estimated depth map and the pixel-level classification results of Mask-RCNN for the pedestrian target, this paper proposed a distance measurement method that calculates the average of the depth values corresponding to all pixels whose depth values are in the peak range of the target region depth histogram. For the vehicle target, this paper proposed a distance measurement method which first fits a plane using

RANSAC, then projects all the pixels from the target to this plane, and finally uses the minimum depth value of these projected points to calculate the distance to the target vehicle. Extensive tests using a public dataset were conducted to assess the results of depth map estimation and real experiments were performed to evaluate the results of the distance measurements. The experimental results using the public dataset proved the superior performance of the proposed depth map estimation method, and the experimental results in real road scenarios confirmed the effectiveness of the distance measurement methods.

Since the accuracy of the proposed distance measurement results depends to some extent on the results of instance segmentation, we plan to combine the depth map and the shape of the target to improve the location precision of masks obtained by instance segmentation and further improve the accuracy of distance measurement. Additionally, in bad visibility conditions caused by illumination, gas particles, dust, fog, etc., the proposed method using images from the visible light sensor does not achieve satisfactory results. Therefore, our research group is studying a completely new method that uses infrared images for distance estimation for ADAS in cases of low visibility.

**Author Contributions:** M.D. and X.J. conceived the research. M.D., Z.Z. and Y.C. developed and implemented the software. M.D. and Z.Z. wrote the paper. M.D., Z.Z., X.J. and Y.C. analyzed the results. All authors have read and agreed to the published version of the manuscript.

**Funding:** This work was partially supported by the National Natural Science Foundation of China (No.61673211, No.U1633105) and by the Fundamental Research Funds for the Central Universities of China (No.NS2020049).

**Conflicts of Interest:** The authors declare they have no conflicts of interest.

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
