# Peer review of "Vision-Based Distance Measurement in Advanced Driving Assistance Systems"

_applsci, doi:10.3390/app10207276_

Round 1

Reviewer 1 Report

The novelty of the authors’ work is that they propose a passive distance measurement system only using monocular vision to complete online absolute distance measurement of vehicle to vehicle or pedestrian to vehicle, frequently employed especially in urban scenarios. However, for achieving this, a single vision point cannot ensure, without an appropriate information processing, distance information with the required accuracy for such type of applications. Therefore, the authors present their research, based on semi-supervised learning platform which is firstly trained with the help of the optical sensor (CCD from the front – looking camera of the vehicle) and the depth labels captured by a LIDAR sensor. Then the algorithm determines the real distances in the two proposed scenarios, that is vehicle-to-vehicle and vehicle-to-pedestrian.

Questions:

  • Did the authors take into consideration the influence of different factors, specific to optical systems, that may affect the precision of distance measurement between front objects, such as: Ambiental illumination (sensitivity of optical sensor may depend on the ambient illumination, angle of lighting, and/or type of street lighting – here I consider the colour temperature) – may these constitute a subject for further research?
  • The optical part of the front – looking camera may also be subject of different influences: aperture influences the depth of field, leading to presenting objects that are not in range as blurred. If there is a variable aperture depending on illumination, the depth of field may also be variable. Therefore, probably the front looking camera should be fixed focus, fixed aperture?
  • The transmittance of the medium between the target object (vehicle, or pedestrian) and the ego automated vehicle is also a factor that may influence the precision of the measurement, due to gas particles, PM, dust, fog etc. How does the method proposed by the authors cope with these adverse manifestations of the environment?
  • There is not very clear what is the delay produced by the processing algorithm and how much could this delay influence the self-driving efficiency of the automated vehicle.

Figure 4 c is very small and difficult to read – is this correctly presented?

Reviewer 2 Report

The paper proposes a depth map estimation method based on monocular vision through semi-supervised learning. My comments are the following.

  1. I'd argue with the statements in lines 46-49. First, the costs of Radar and ultrasonic are low; only the LIDAR is expensive. Second, neglecting these sensors and using only vision-based systems poses problems on wrong visibility scenarios. I'd suggest rewriting these thoughts as complementary sensors.
  2. The literature review is shallow. The authors should give a broader literature review.
  3. Also, discussion of other approaches contain phrases like "However, the shortcoming of these methods is also very obvious." (Line 120) Though it is a stylistic issue, be more detailed and more generous on writing a literature review. 
  4. Also, just writing style, but "As we all know" does not look good.
  5. There are three appearances of "In our opinion..." It is ok to write subjective thoughts, though these opinions can be proven or cited from somewhere else in some cases.
  6. Figure 3: Part (4) is small. Work on it. 
  7. Figure 4: Part (c) is small. Work on it.
  8. Also, if there are subfigures, use one paradigm, either using (1)-(2)... or (a)-(b)...
  9. It can be my fault, but the complete toolchain is not understandable. What is the point where stereo vision separates from mono? Please clarify it in the paper. My main concern is this in the review. Please make an effort in it.
  10. Please clearly distinguish your algorithm from the ones that can be found in the literature.
  11. It is also not clear why are there separate branches for pedestrians and vehicles? (Like Figure 5.) And what about other objects?
  12. Regarding vision-based solutions, it is always beneficial to write about the limitations. I understand that it is hard to perform real tests, though, in the KITTI dataset, there are images with bad visibility conditions. The authors should provide some examples of how their algorithm works with them.

Reviewer 3 Report

In line 383 the authors mention that they use 200 images of KITTI as the test samples. Is this from KITTI"test" dataset (no ground truth available) or from the annotated training set? Why do you only use 200 images? It would be much better if the authors can provide their results on the official KITTI test split.

Round 2

Reviewer 2 Report

Thank you, may questions and concerns are covered in the revised manuscript, I don't have any other questions.